# Detection of LUAD-Associated Genes Using Wasserstein Distance in Multiomics Feature Selection

**DOI:** 10.3390/bioengineering12070694

**Published:** 2025-06-25

**Authors:** Shaofei Zhao, Siming Huang, Lingli Yang, Weiyu Zhou, Kexuan Li, Shige Wang

**Affiliations:** 1Statistical Sciences, AbbVie, Florham Park, NJ 07932, USA; 2Department of Mathematics and Statistics, Binghamton University, Binghamton, NY 13902, USA; shuang87@binghamton.edu; 3Department of Biostatistics, Johns Hopkins Bloomberg School of Public Health, Baltimore, MD 21205, USA; lyang127@jh.edu; 4Department of Statistics, George Mason University, Fairfax, VA 22030, USA; weiyu_zhou@vrtx.com; 5Global Biometrics and Data Sciences, Bristol Myers Squibb, Cambridge, MA 02139, USA; kli77@binghamton.edu; 6College of Engineering, Northeastern University, Boston, MA 02115, USA

**Keywords:** feature selection, multiomics data, LUAD, tumor mutational burden, Wasserstein distance

## Abstract

Lung adenocarcinoma (LUAD) is characterized by substantial genetic heterogeneity, making it challenging to identify reliable biomarkers for diagnosis and treatment. Tumor mutational burden (TMB) is widely recognized as a predictive biomarker due to its association with immune response and treatment efficacy. In this study, we take a different approach by treating TMB as a response variable to uncover its genetic drivers using multiomics data. We conducted a thorough evaluation of recent feature selection methods through extensive simulations and identified three top-performing approaches: projection correlation screening (PC-Screen), distance correlation sure independence screening (DC-SIS), and Wasserstein distance-based screening (WD-Screen). Unlike traditional approaches that rely on simple statistical tests or dataset splitting for validation, we adopt a method-based validation strategy, selecting top-ranked features from each method and identifying consistently selected genes across all three. Using The Cancer Genome Atlas (TCGA) dataset, we integrated copy number alteration (CNA), mRNA expression, and DNA methylation data as predictors and applied our selected methods. In the two-platform analysis (mRNA + CNA), we identified 13 key genes, including both previously reported LUAD-associated genes (*CCNG1, CKAP2L, HSD17B4, SHROOM1, TIGD6*, and *TMEM173*) and novel candidates (*DTWD2, FLJ33630, NME5, NUDT12, PCBD2, REEP5*, and *SLC22A5*). Expanding to a three-platform analysis (mRNA + CNA + methylation) further refined our findings, with *PCBD2* and *TMEM173* emerging as the robust candidates. These results highlight the complexity of multiomics integration and the need for advanced feature selection techniques to uncover biologically meaningful patterns. Our multiomics strategy and robust selection approach provide insights into the genetic determinants of TMB, offering potential biomarkers for targeted LUAD therapies and demonstrating the power of Wasserstein distance-based feature selection in complex genomic analysis.

## 1. Introduction

Lung adenocarcinoma (LUAD), a predominant subtype of non-small cell lung cancer (NSCLC), accounts for nearly 40% of all lung cancer cases worldwide, making it a critical focus in oncology research. According to the Global Cancer Observatory (GLOBOCAN), lung cancer remains the leading cause of cancer-related deaths, with over 2 million new cases and approximately 1.8 million deaths reported annually [1], and LUAD comprises the majority of these cases. The prognosis for LUAD patients remains poor, with a five-year survival rate below 20%, particularly due to late-stage diagnoses when treatment options are limited.

Recent advancements in multiomics technologies have provided a deeper understanding of the molecular landscape of LUAD, and several studies have focused on multiomics approaches and machine learning techniques to extract highly related genes. For example, using feature selection frameworks with mutual information and random forest, researchers found a consensus set of twelve genes with significant diagnostic potential, which could differentiate LUAD from normal samples with high accuracy [2]. In another study, Guo et al. applied a deep learning-based multiomics integration strategy to identify subtype-specific biomarkers and improve LUAD classification, demonstrating the utility of neural network-based methods for biomarker discovery [3]. However, despite these advancements, high genetic and molecular heterogeneity in LUAD continues to limit the applicability of existing biomarkers for early and personalized diagnostics.

The analysis of multiomics data presents significant challenges due to its high dimensionality, with tens of thousands of genes but only a few hundred subjects and the integration of multiple data platforms. This structure introduces a high degree of noise and creates an array-like predictor structure, where for *n* subjects, the predictors not only have the regular high dimensionality *p* but also an additional dimension *d* associated with *d* different platforms. This complexity makes traditional analysis methods ineffective and sometimes even incapable of handling predictors with such a multi-layer structure. As a result, effective feature selection is crucial for detecting signals in this noisy environment.

Feature selection, or feature screening, has long been a hot topic in statistical and machine learning research, particularly due to its critical role in managing high-dimensional and complex data. Fan and Lv’s introduction of sure independent screening (SIS) was a pivotal development, demonstrating that SIS could reliably identify all true predictors in sufficiently large samples, hence the term “sure” screening [4]. Their work inspired further advancements in sure screening, with methods like distance correlation-based sure independence screening (DC-SIS) [5], projection correlation-based screening (PC-Screen) [6], stable correlation screening (SC-SIS) [7], and multivariate rank distance correlation-based sure independence screening (MrDc-SIS) [8]. For a comprehensive review and comparison of these robust screening techniques, Zhao and Fu [8] offers detailed insights. Recently, Zhao et al. [9] utilized a model-free and distribution-free screening method, MrDcGene, on The Cancer Genome Atlas Lung adenocarcinoma (TCGA-LUAD) dataset. This method effectively confirmed known biomarkers and identified new gene candidates associated with LUAD, showcasing the potential of advanced sure screening methods in handling the intrinsic complexity of multiomics data.

Building on recent advancements in dependence measures, we noted a new approach using the Wasserstein distance as a dependence metric, as introduced in the work by Mordant and Segers [10]. The Wasserstein distance is a method for comparing probability distributions by quantifying the minimal cost of transforming one distribution into another, where the cost is defined in terms of how much “mass” must be moved and how far. This concept, rooted in optimal transport theory, enables a flexible and geometrically meaningful comparison of complex data structures.

The Wasserstein distance offers several advantages. First, it remains stable under transformations like rotation and monotonic changes, making it robust in noisy and high-dimensional data. Second, unlike some traditional measures, it does not rely on assumptions about linearity or Gaussian distributions, which allows it to work effectively in model-free and distribution-free settings, making it a versatile choice for complex mulit-omics data. Finally, the Wasserstein distance measures the dependence by minimizing the “transport cost” between distributions, grounded in optimal transport theory, providing a reliable and theoretically sound way to measure associations.

These properties make the Wasserstein distance particularly well-suited for high-dimensional multiomics datasets like the TCGA-LUAD, where both complexity and noise are prevalent. In this paper, we aim to thoroughly test and compare the performance of the Wasserstein distance against other feature selection methods through extensive simulation studies. Following these simulations, we apply the Wasserstein distance-based approach to the real TCGA-LUAD dataset, which we downloaded through cBioPortal [11], an open-access platform for exploring multidimensional cancer genomics data, to evaluate its effectiveness in identifying meaningful gene associations for LUAD.

In both the simulation and real data analyses, our primary objective is to identify important genomic features (genes) that are strongly associated with a continuous outcome. For the synthetic data experiments, the features are high-dimensional vectors representing simulated gene expression profiles or multi-platform measurements, and the response variable is either a continuous scalar (in Studies 1 and 2) or a multivariate vector (in Studies 3 and 4), generated based on linear or interaction-based models. In the real data analyses, the features are gene-level measurements from multiomics platforms (CNA, mRNA expression, and DNA methylation), and the response variable is the tumor mutational burden (TMB), a continuous clinical metric. Our feature selection approach ranks genes based on their marginal dependence with the response variable, with the goal of selecting a small subset of genes most informative of TMB variation.

## 2. Results

To provide an overview of our experimental design, we summarize all simulation studies and real data analyses in Table 1. In all experiments, the task is formulated as a regression-based feature screening problem, where the goal is to rank and select predictors (genes) that exhibit strong marginal association with a continuous outcome. The simulation studies were constructed to systematically evaluate the performance of various feature selection methods under increasingly complex scenarios. Simulation 1 serves as a benchmark case using ideal normal predictors with a linear response, ensuring correct implementation and baseline performance. Simulation 2 introduces skewed non-Gaussian predictors to mimic real-world distributional characteristics observed in genomic data. Simulation 3 extends this to a multiomics structure by incorporating three types of predictors: copy number alteration (CNA), mRNA expression, and methylation, along with a multivariate response, closely reflecting the structure of the actual TCGA-LUAD dataset. Simulation 4 introduces interaction effects among predictors to assess whether the methods can still identify relevant features when marginal associations are weak or absent.

We also conduct two real data analyses using the TCGA-LUAD dataset with tumor mutational burden (TMB) as the response. The first uses a two-platform design (CNA and mRNA), while the second incorporates all three omics platforms (CNA, mRNA, and methylation). For each setting, we apply the three most robust methods identified in our simulations, PC-Screen, DC-SIS, and WD-Screen, and focus on genes consistently selected in all three methods. This strategy not only validates the simulation findings but also demonstrates the practical utility of these methods in uncovering biologically meaningful biomarkers in complex high-dimensional genomic data.

### 2.1. Comparative Analysis of Feature Selection Methods: Simulation Studies

In our comparison, we evaluated ten popular feature selection methods, including our Wasserstein distance-based screening (WD-Screen). The other methods are as follows; for details, please refer to Zhao and Fu [8]:Sure Independence Screening (SIS) [4]—uses Pearson correlation as the dependence measure between each predictor and response variable.Sure Independence Ranking and Screening (SIRS) [12]—uses Pearson correlation between each predictor and the rank of response variable.Robust Rank Correlation Screening (RRCS) [13]—uses Kendall’s τ as the dependence measure between each predictor and the response variable.Distance Correlation-based Sure Independence Screening (DC-SIS) [5]—uses distance correlation as the dependence measure between each predictor and the response variable.Robust Distance Correlation Sure Independence Screening (DC-RoSIS) [14]—uses distance correlation between each predictor and the rank of response variable.Multivariate Rank-based Distance Correlation Sure Independence Screening (MrDc-SIS) [8]—uses distance correlation between the multivariate rank of predictors and response variables.Stable Correlation-based Sure Independence Screening (SC-SIS) [7]—uses a different weight function in the distance correlation as the dependence measure between each predictor and response variable.Projection Correlation-based Screening (PC-Screen) [6]—uses projection correlation as the dependence measure between each predictor and response variable. Projection correlation is a model-free dependence measure that captures both linear and nonlinear associations by projecting high-dimensional data onto lower-dimensional subspaces and measuring the correlation between these projections. It is particularly effective in high-dimensional settings where complex structures and interactions may exist, and it retains sensitivity to a wide range of dependence types without relying on strong parametric assumptions.Ball Correlation-based Sure Independence Screening (BCor-SIS) [15]—uses ball correlation as the dependence measure between each predictor and response variable. Ball correlation is a nonparametric and model-free dependence measure that can detect both linear and nonlinear associations, including complex relationships where traditional correlation measures fail. It is based on the idea of comparing distances between observations within “balls” (or neighborhoods) in the feature and response spaces, making it well-suited for high-dimensional and non-Euclidean data structures. This flexibility allows BCor-SIS to perform robustly in settings where standard assumptions (e.g., linearity or Gaussianity) do not hold.

#### 2.1.1. Simulation 1: Benchmarking and Validation of Feature Selection Methods

To establish a benchmark, allow comparison with prior studies, and verify the correctness of our implementation for each method, we replicated a similar simulation setup as used in previous papers [4,5,8]. In this study, we generated data with n=200 observations and p=2000 predictors. The predictors $X_{n\times p}$ were drawn from a multivariate normal distribution with a zero mean and autoregressive order-1 (AR (1)) covariance structure, where the covariance matrix $\Sigma_{p\times p}={\left[\sigma_{ij}\right]}_{p\times p}$, and $\sigma_{ij}=0.5^{\left|i-j\right|}$ for i,j=1,2,…,p. The response variable $y_{n\times 1}$ was constructed by$$y=\beta_{1}X_{1}+\beta_{2}X_{2}+\beta_{3}X_{12}+\beta_{4}X_{13}+\epsilon ,$$
where $\epsilon_{n\times 1}$ is standard normal and $\beta_{1,2,3,4}∼Uniform\left(2,5\right)$.

We repeated the simulation 200 times, and following Li et al. [5], we utilized three criteria to assess the performance of the feature selection methods:S: The minimum model size to include all true predictors. We draw a box plot of S. The smaller the S, the better the performance.$P_{s}$: The individual success rate of selecting a single true predictor within a predetermined cutoff across 200 replicates. The larger the $P_{s}$, the better the performance.$P_{a}$: The success rate of selecting all true predictors within a predetermined cutoff across 200 replicates. The larger the $P_{a}$, the better the performance.

We set the predetermined cutoff as [n/log(n)], where [a] stands for the integer part of *a*, to be consistent with Li et al. [5]. For n=200, the cutoff was s=[n/log(n)]=37. Aside from the three criteria above, we also created a box plot for the rank of each true predictor. As a smaller rank indicates a more important predictor, ideally, the true predictors should be ranked at the top. This visualization can further highlight each method’s effectiveness in identifying the true predictors.

All methods performed well under these simple settings. For most, the selection size S remained below 10 (Figure 1), with only occasional instances where S exceeded 20 in SC-SIS. Figure 2 highlights that each true predictor was ranked near the top. While SC-SIS and BCor-SIS occasionally ranked true predictors above 10, the overall performance across methods was strong. Further, Table 2 shows that both $P_{s}$ and $P_{a}$ are close to 1, confirming the accuracy of our implementations and their effectiveness in this scenario.

#### 2.1.2. Simulation 2: Evaluating Performance with Non-Normal Predictor Distributions

In this study, we changed the distribution of the predictors, $X_{i}$, to follow an independent and identically distributed (i.i.d.) power function distribution, which is the inverse of the Pareto distribution with probability density function $f\left(x;a\right)=ax^{a-1}$ for 0<x<1 and a>0. In our simulation, we chose the parameter a=5, A typical histogram of the power distribution (Figure 3) below illustrates how it more closely resembles real-world data compared to the normal distribution used in Simulation 1. In Figure 4, we observe that, although the real data has been standardized, it remains left-skewed rather than symmetric, as seen in normally distributed data. In this sense, the power distribution provides a more realistic representation of the skewed nature of real data. To further increase the difficulty of the test, the coefficients $\beta_{1,2,3,4}∼Uniform\left(1,2\right)$, which were smaller than in Simulation 1. All other settings remained the same as in Simulation 1. We repeated the simulation 200 times visualizing S and the performance of each individual true predictor. Additionally, we report $P_{s}$ and $P_{a}$ to provide a comprehensive evaluation of each method.

When the predictors followed a power distribution, the performance of all methods declined. The median S ranged from approximately 250 to 750 (Figure 5), significantly higher than in Simulation 1. SIS and WD-Screen performed slightly better than the other methods, with a median S around 250. SC-SIS and BCor-SIS performed worse, with a median S around 750. Breaking down by true predictor, SIS and WD-Screen excelled in $X_{1}$ and are comparable in other predictors, while SC-SIS an BCor-SIS consistently performed poorly, displaying larger variability in the boxplot for all true predictors (Figure 6). Table 3 supports our observations: SIS and WD-Screen have the highest $P_{a}$ among all methods, while SC-SIS and BCor-SIS have the lowest $P_{a}$.

#### 2.1.3. Simulation 3: Simulating Multiomics and Multi-Endpoint Data Structures

To more closely simulate the structure of multiomics data, we adapted the settings based on a TCGA-LUAD simulation from Zhao et al. [9]. In this study, we used n=200 observations, with p=2000 predictor array and a q=10-dimensional response vector. Each predictor was represented as a three-dimensional vector to mimic the multiomics settings, where data are collected across three different platforms (e.g., copy number variation, RNAseq, etc.). The 10-dimensional response vector, in turn, reflects real-world situations where multiple clinical or biological endpoints are analyzed simultaneously.

The three platforms were generated as follows:Platform 1 ($U=[U_{1},U_{2},\ldots ,U_{p}]$): multivariate Pareto distributed with shape $a_{U}=10$ and mode $m_{U}=1$.Platform 2 ($V=[V_{1},V_{2},\ldots ,V_{p}]$): multivariate power distributed with parameter $a_{V}=5$.Platform 3 ($W=[W_{1},W_{2},\ldots ,W_{p}]$): multivariate power distributed with parameter $a_{W}=5$.

Each platform has a shared covariance structure $\Sigma_{p\times p}={\left[\sigma_{ij}\right]}_{p\times p}$, where $\sigma_{ij}=0.5^{\left|i-j\right|}$, and the predictor array $X_{3\times n\times p}=\left[X_{1},X_{2},\ldots ,X_{p}\right]$ is constructed by stacking these platforms: $X_{j}=\left[U_{j},V_{j},W_{j}\right]$, ∀j=1,2,…,p.

The response vector is connected with the predictors as follows:For k=1,2,31.Randomly select indices $id_{1,2,3,4}$ from {1,2,3} to represent the true platforms connected with the response.2.$Y_{k}\left[i\right]=\beta_{1}X_{id_{1},i,2}+\beta_{2}X_{id_{2},i,3}+\beta_{3}X_{id_{3},i,101}+\beta_{4}X_{id_{4},i,102}+\epsilon \left[i\right]$, ∀i=1,2,…,n, where $\beta_{1,2,3,4}∼Uniform\left(1,2\right)$ and ϵ∼N(0,1).For k=4,5,…,10, $Y_{k}∼Power\left(5\right)$.

Since SIS, SIRS, RRCS, and DC-RoSIS cannot handle multivariate response and predictors, we only compared the performance of the other six methods.

Under this complex setting, PC-Screen, DC-SIS, and WD-Screen demonstrate comparable performance, consistently outperforming other methods in both S (see Figure 7) and ranking of individual true predictors (Figure 8). Actually, these three methods perform even better than in the single predictor settings in Simulation 2. SC-SIS and BCor-SIS show similar results to each other, while MrDc-SIS performs the worst. Table 4 further highlights these findings: with a relatively small model size (37), PC-Screen and DC-SIS have a higher than 50% chance of selecting all four true predictors, and WD-Screen achieves a nearly 50% selection rate for $P_{a}$ as well.

#### 2.1.4. Simulation 4: Assessing Feature Selection with Interaction Effects

In this study, we introduced an interaction term to simulate real-world scenarios where certain genes may not have an individual effect on the disease but may influence it in combination with others. Most settings remained the same as in Simulation 3, but the active response for k=1,2,3 was$$Y_{k}\left[i\right]=\beta_{1}X_{id_{1},i,2}\times X_{id_{2},i,3}+\beta_{2}X_{id_{3},i,101}\times X_{id_{4},i,102}+\epsilon \left[i\right],\forall i=1,2,\ldots ,n,$$
where $\beta_{1,2}∼Uniform\left(1,2\right)$ and ϵ∼N(0,1).

Traditional feature selection methods often struggle in such settings because they primarily assess marginal effects, failing to capture interactions between predictors. However, despite the absence of direct associations, PC-Screen, DC-SIS, and WD-Screen still outperform other methods, achieving a success selection rate of approximately 0.8 for each true predictor (Table 5). Their performance in this interaction setting remains comparable (see Figure 9 and Figure 10) to Simulation 3, where predictors had direct effects on the response, further reinforcing their robustness. This suggests that while marginal screening methods do not explicitly model interactions, these advanced methods can still effectively capture features involved in interactions by leveraging nonlinear and dependence-based selection criteria. Given that gene–gene interactions play a critical role in complex diseases such as LUAD, these results highlight the potential of our approach in identifying biologically meaningful relationships within high-dimensional multiomics data.

### 2.2. Feature Selection Methods on Real-World Data: A TCGA-LUAD Case Study

We obtained the TCGA-LUAD data from cBioPortal [11] and selected the nonsynonymous tumor mutational burden (TMB) from clinical sample data as our response variable. TMB, particularly nonsynonymous TMB, is an important biomarker in cancer research, measuring the total number of somatic nonsynonymous mutations per megabase in tumor cells, and it can vary widely across and within cancer types. High TMB levels increase the production of neoantigens, which may be recognized by the immune system, potentially enhancing the efficacy of immunotherapy. Recently, studies have shown that TMB is associated with clinical outcomes in multiple cancers, including melanoma, non-small-cell lung cancer, and colorectal cancer. Evidence suggests that a high TMB can effectively predict objective response rates and progression-free survival, making it a valuable indicator in assessing immunotherapy outcomes [16,17].

In our study, we treated TMB as the response variable and sought to understand its association with genetic variations by integrating data from multiomics platforms. Specifically, we used data on copy number alteration (CNA), DNA methylation, and mRNA expression as predictors. Each of these platforms provides unique insights: CNA data reflect gene amplification or deletion, methylation data reveal epigenetic changes that may influence gene expression, and mRNA expression levels indicate gene activity. By combining information from these platforms, we aimed to identify genes whose variations correlate strongly with TMB, uncovering potential drivers of mutational burden in LUAD. This multiomics approach allows us to explore complex cross-platform interactions and their influence on TMB, which could ultimately inform targeted strategies for immunotherapy in lung cancer.

From our simulation studies, we observed that PC-Screen, DC-SIS, and WD-Screen consistently outperform other feature selection methods, showing reliable results across diverse scenarios. For the real data analysis, we applied these three methods to our dataset, each offering a distinct approach and methodology to feature selection. Instead of using a traditional training–testing split, we adopted a more robust selection strategy that used the full information of the real data: for each method, we selected the top [n/log(n)] features to form a selection set; then, we chose the intersection of the three selection sets as our final selection set. By focusing on features that were independently identified by all three top-performing methods, we aimed to enhance the robustness and reliability of our final selection, ensuring that the chosen features are more likely to have genuine associations with TMB.

#### 2.2.1. Two-Platform Study

In our first study, we combined CNA and mRNA data, using the files “data_mrna_seq_ v2_rsem_zscores_ref_all_samples.txt” for mRNA and “data_cna.txt” for CNA. The original CNA data contained 230 subjects with 23,423 genes; the data were discrete, where −2 represents homozygous deletion, −1 represents hemizygous deletion, 0 reprents no change, 1 represents gain, and 2 represents high level amplification. The mRNA data contained 230 subjects with 20,466 genes; they were continuous, and they were log-transformed and standardized over the expression distribution of all samples. After removing the duplicates and genes missing from one platform, there were 18,674 common genes across 230 subjects in these two platforms. Our predictor X was a 2×230×18,674 array, and our response Y was a 230×1 vector.

The three methods had 13 genes in common among the top [230/log(230)]=42 selected genes: *CCNG1, CKAP2L, DTWD2, FLJ33630, HSD17B4, NME5, NUDT12, PCBD2, REEP5, SHROOM1, SLC22A5, TIGD6*, and *TMEM173*. Below is a visulization of the relationship between TMB and the CNA and mRNA of the 13 genes. The results also show that these three methods can handle categorical variables and continuous variables together in one predictor vector.

Our study not only confirmed previously reported findings, but also uncovered novel insights into LUAD genetics. Among the identified genes, several have been validated by prior research. *CCNG1* has been highlighted for its involvement in MYC-mediated regulation, influencing cell division and proliferation in LUAD [18]. *CKAP2L* is associated with a poor prognosis, with high expression associated with advanced cancer stages and metastases [19]. *HSD17B4* plays a role in immune regulation and tumor progression, which is positively correlated with immune cell infiltration in LUAD [20]. *SHROOM1* has been identified as a somatic hotspot mutation in major cancers, including LUAD [21]. *TIGD6* contributes to genomic instability in LUAD by facilitating copy number alterations (CNAs) and is associated with poorer survival [22]. *TMEM173*, a key regulator of immune responses, shows reduced expression in LUAD tissues, with higher expression correlating with improved overall survival [23]. These findings reinforce our methodology by independently corroborating prior discoveries using a different dataset.

In addition to confirming these previously reported genes, our study identified several novel genes in LUAD that have not been extensively documented. For example, *DTWD2* may influence genomic stability and cancer pathways [24]. *FLJ33630*, a long non-coding RNA, is emerging as a potential biomarker due to its role in regulating gene expression and cancer progression [25]. *NME5* functions as a tumor suppressor, with mutations linked to aggressive cancer phenotypes and genomic instability [26]. *NUDT12*, part of the Nudix family, impacts the regulation of the nucleotide pool, affecting DNA stability and repair in cancer [27]. *PCBD2*, though previously studied in rectal cancer, may have tumor-suppressive potential in LUAD [28]. *REEP5* is associated with pathways like PI3K-Akt signaling and oxidative phosphorylation, crucial for cancer growth and survival [29]. Finally, *SLC22A5*, implicated in pancreatic and breast cancers, may contribute to LUAD progression through metabolic regulation [30].

Figure 11 shows a distinct trend in the CNA of these 13 genes. For 12 of them, the TMB decreases as the copy number shifts from deletion to neutral and then to gain or amplification. However, *CKAP2L* shows an opposite pattern, where TMB positively correlates with its CNA. This observation is further supported by the scatter plot of mRNA expression versus TMB, where the expression of *CKAP2L* increases as TMB increases, while the expression level of all other genes decreases. The negative relationship between TMB and CNA may be due to the impact of CNA on genome stability and cell proliferation. High CNAs, especially amplifications, are often associated with increased cellular stability and reduced mutation rates. Cells with amplified oncogenes may activate robust DNA repair mechanisms to maintain genomic integrity, therefore reducing the accumulation of additional mutations and resulting in low TMB [31]. Conversely, gene deletions can lead to the loss of tumor suppressor genes, contributing to genomic instability and potentially allowing for a higher mutation burden. The relationship between CNA and gene expression also supports this observation, as genes with higher copy numbers tend to maintain higher expression levels and cellular stability, further reducing opportunities for mutation accumulation [32].

We also plot the gene expression at each cancer stage in Figure 12. Among the 230 subjects, there were 12 subjects with cancer stage missing. As shown in the strip plot (orange dots in Figure 12) and Table 6, roughly half of the subjects were in Stage I. Overall, we do not see a clear trend in gene expression across cancer stages. However, for most of the 13 genes (except *CKAP2L*), the expression decreases at Stage IIA and then quickly increases at Stage IIB. This pattern may suggest that Stage IIA in LUAD exhibits distinct molecular characteristics compared to other stages, indicating that there may be a potential differences in disease progression. This finding could also support exploring tailored treatment strategies specifically for Stage IIA patients, as their gene expression profiles may reflect unique therapeutic needs. There is another observation in Stage IV: although there are only nine patients, the gene expressions of *CCNG1, CKAP2L, DTWD2, HSD17B4, PCBD2* were spread more widely than in previous stages. This increased variability suggests greater heterogeneity among subjects at later stages of LUAD, potentially reflecting more diverse underlying molecular characteristics. Although Stage IV is often considered incurable, this variation may open possibilities for personalized treatment strategies to enhance the quality of life, providing tailored approaches that better address individual needs at a later stage.

#### 2.2.2. Three-Platform Study

Aside from the CNA and mRNA data, we also investigated the DNA methylation data in file “data_methylation_hm450.txt”, which were continuous with 185 samples and 16,240 genes. After removing the duplicates and missings, there were 14,399 common genes and 185 samples among all three platforms. Our predictor X was a 3×185× 14,399 array, and our response was a 185×1 vector.

We again applied PC-Screen, DC-SIS, and WD-Screen to the data and selected the top [185/log(185)]=35 genes in each method: the intersection of all three selection sets yielded two genes: *PCBD2* and *TMEM173*.

From Figure 13, we can see the CNA, gene expression, and DNA methylation with TMB of these two genes, and they share a similar trend.

## 3. Discussion

In this study, we applied a Wasserstein distance-based feature selection method to identify genomic features associated with tumor mutational burden (TMB) in LUAD using multiomics data from TCGA. Instead of using traditional validation methods, such as splitting the dataset into training and testing sets or comparing results from different datasets, we adopted a method-based validation strategy. Specifically, we selected the top [n/log(n)] features using three of the most robust feature selection methods in our simulations, PC-Screen, DC-SIS, and WD-Screen, an then identified the genes commonly selected across all three methods.

Using this approach, our initial analysis on two-platform data (mRNA expression and CNA) identified 13 key genes, including *CCNG1, CKAP2L, DTWD2, FLJ33630, HSD17B4, NME5, NUDT12, PCBD2, REEP5, SHROOM1, SLC22A5, TIGD6*, and *TMEM173*. Some of these genes have been previously identified in LUAD studies using different analytical methods, for example *CCNG1, CKAP2L, HSD17B4, SHROOM1, TIGD6* and *TMEM173*, further validating our approach. Meanwhile, several of these genes, such as *DTWD2, FLJ33630, NME5, NUDT12, PCBD2, REEP5* and *SLC22A5*, have not been extensively studied in the context of LUAD, representing novel findings that could provide new insights into LUAD biology.

Expanding our analysis to a three-platform dataset (incorporating DNA methylation in addition to mRNA and CNA) further refined our findings, with *PCBD2* and *TMEM173* emerging as strong candidates. These two genes were consistently selected across both analyses (Two-platform and Three-platform) and all three feature selection methods, reinforcing their potential significance in LUAD.

Our results demonstrate that incorporating multiomics data and advanced dependence measures improves feature selection, capturing nonlinear associations that may be overlooked by traditional methods. The identification of these genes provides valuable insights into LUAD’s genomic landscape and its relationship with TMB, a key biomarker in immunotherapy response.

Multiomics data analysis has gained increasing attention in cancer research; yet, many existing studies rely on relatively simple statistical methods for feature selection. Traditional approaches often use *t*-tests, ANOVA, or fold-change analyses to identify differentially expressed genes across conditions [33,34]. While these methods are easy to implement, they do not account for complex nonlinear interactions within multiomics data, potentially overlooking biologically significant patterns. Additionally, some studies apply penalized regression models, such as LASSO or Elastic Net, which assume sparsity but can struggle in ultra-high-dimensional settings where the number of features far exceeds the number of samples [4]. Our study distinguishes itself by implementing advanced feature selection methods—namely, PC-Screen, DC-SIS, and WD-Screen—that possess the sure screening property. This property ensures that, given a sufficiently large sample size, the probability of selecting truly important genes approaches one. Consequently, our approach not only re-identifies genes previously reported in the literature but also uncovers novel genes that have not been extensively discussed. This capability highlights the strength of our methods in detecting subtle hard-to-find patterns within complex datasets.

In addition to the above, our study introduces several methodological advancements that enhance the reliability and efficiency of multiomics feature selection.

No Need for Extensive Preprocessing or Dimensionality Reduction: Many multiomics studies require extensive preprocessing steps, such as manually filtering genes with low variability or reducing the feature space due to computational constraints. These steps can introduce subjective biases and may exclude biologically relevant features. In contrast, our approach utilizes marginal screening methods, which assess each predictor individually, making it naturally scalable to high-dimensional data. This allows us to analyze millions of genes without the need for dimensionality reduction, preserving the integrity of the original dataset.Robustness Against Collinearity in Genomic Data: Gene expression and other omics features often exhibit strong collinearity, which can be problematic for many traditional feature selection methods. Our marginal screening-based approach eliminates this concern, as each feature is evaluated independently for its association with the response variable. This ensures that important genes are not overlooked due to multicollinearity issues, leading to a more reliable selection process.Validation through Multiple Methods Instead of Data Splitting: A common validation strategy in machine learning and statistical modeling is to split the dataset into training and testing sets or use external datasets for replication. However, in multiomics research, where sample sizes are often limited, splitting the dataset further reduces the statistical power. Additionally, different datasets may have variations in data processing, measurement techniques, and platform biases, making direct comparisons challenging. Instead of relying on dataset splitting, we validate our findings by identifying genes consistently selected by three independent robust feature selection methods, which provides a stronger methodological validation than simply finding overlap between two datasets.

While our study presents a robust feature selection framework for multiomics LUAD data, several challenges remain that offer opportunities for future improvement. For example, our results highlight that as we expanded from two-platform (mRNA + CNA) to three-platform (mRNA + CNA + methylation) data, the number of common genes selected by all three methods decreased significantly. This suggests that as additional omics layers are incorporated, the complexity and noisiness of the data increase, making it harder to identify robust biomarkers, even when using advanced and statistically rigorous methodologies. This finding underscores the need for both more robust and accurate feature selection methods that can effectively integrate information across multiple omics layers while filtering out noise, as well as higher-quality standardized multiomics datasets. Discrepancies in data collection, normalization, and platform-specific biases introduce inconsistencies that complicate analysis. Future research should focus on improving data preprocessing pipelines and developing novel integration strategies to enhance the signal-to-noise ratio in multiomics studies.

Despite these challenges, our study provides a strong statistical foundation for multiomics feature selection and highlights both biologically relevant findings and methodological insights. Future work should explore interaction-aware feature selection and strategies to enhance data quality, ultimately improving our ability to identify reliable genomic biomarkers in LUAD and other cancers.

## 4. Materials and Methods

Among the various feature screening methods we evaluated, some are relatively straightforward to implement and are omitted here for brevity. Specifically, SIS, SIRS, and RRCS rely on simple correlation measures such as Pearson correlation or Kendall’s τ between each predictor and the response variable. Furthermore, DC-RoSIS, MrDc-SIS, and SC-SIS are variations or extensions of the DC-SIS framework that use ranked or weighted versions of distance correlation. As our focus is on evaluating robust and representative methods, we briefly summarize the three main approaches used in our real data analyses: DC-SIS, PC-Screen, and WD-Screen.

### 4.1. DC-SIS

DC-SIS ranks predictors based on their distance correlation with the response variable. Distance correlation is a nonparametric dependence measure that captures both linear and nonlinear associations. The empirical distance correlation between a predictor $X_{j}$ and a response variable *Y* is defined as$$\mathrm{dCor}\left(X_{j},Y\right)=\frac{\mathrm{dCov}(X_{j},Y)}{\sqrt{\mathrm{dCov}\left(X_{j},X_{j}\right)\cdot \mathrm{dCov}\left(Y,Y\right)}},$$
where dCov(·,·) denotes the distance covariance between two variables. Specifically, given *n* observations ${\left\{\left(X_{ij},Y_{i}\right)\right\}}_{i=1}^{n}$, the empirical distance correlation between predictor $X_{j}$ and response *Y* is computed as$${\mathrm{dCor}}^{2}\left(X_{j},Y\right)=\frac{\frac{1}{n^{2}}\sum_{k=1}^{n}\sum_{\ell =1}^{n}A_{k\ell }B_{k\ell }}{\sqrt{\left(\frac{1}{n^{2}}\sum_{k=1}^{n}\sum_{\ell =1}^{n}A_{k\ell }^{2}\right)\left(\frac{1}{n^{2}}\sum_{k=1}^{n}\sum_{\ell =1}^{n}B_{k\ell }^{2}\right)}},$$
where $A_{k\ell }$ and $B_{k\ell }$ are the elements of the “double-centered distance matrices” for $X_{j}$ and *Y*, respectively:$$A_{k\ell }=a_{k\ell }-{\bar{a}}_{k\cdot }-{\bar{a}}_{\cdot \ell }+{\bar{a}}_{\cdot \cdot },\;B_{k\ell }=b_{k\ell }-{\bar{b}}_{k\cdot }-{\bar{b}}_{\cdot \ell }+{\bar{b}}_{\cdot \cdot }$$$$a_{k\ell }=|X_{kj}-X_{\ell j}|,\;b_{k\ell }=\left|Y_{k}-Y_{\ell }\right|,$$
and the dot notations represent averages across rows, columns, or the whole matrix:$${\bar{a}}_{k\cdot }=\frac{1}{n}\sum_{m=1}^{n}a_{km},\;{\bar{a}}_{\cdot \ell }=\frac{1}{n}\sum_{m=1}^{n}a_{m\ell },\;{\bar{a}}_{\cdot \cdot }=\frac{1}{n^{2}}\sum_{k,\ell =1}^{n}a_{k\ell }$$(similarly for ${\bar{b}}_{k\cdot },{\bar{b}}_{\cdot \ell },{\bar{b}}_{\cdot \cdot }$).

Each predictor $X_{j}$ is ranked by $\mathrm{dCor}(X_{j},Y)$, and the top [n/log(n)] predictors are retained.

### 4.2. PC-Screen

PC-Screen uses projection correlation to assess the dependence between predictors and response. It projects both $X_{j}$ and *Y* onto lower-dimensional subspaces and computes the maximum correlation between projections:$$\mathrm{pCor}\left(X_{j},Y\right)=\underset{∥u∥=1,∥v∥=1}{\sup}\mathrm{Corr}\left(u^{⊤}X_{j},v^{⊤}Y\right),$$
where $u\in R^{p}$ and $v\in R^{q}$ are unit vectors, and Corr(·,·) denotes the Pearson correlation. In practice, *u* and *v* are estimated by optimizing over a finite set of projection directions. Each predictor is scored using $\mathrm{pCor}(X_{j},Y)$ and ranked accordingly.

### 4.3. WD-Screen

Direct computation of the Wasserstein distance between empirical distributions can be computationally expensive. To address this, we adopt the efficient formulation from Proposition 5 of Mordant and Segers [10], which expresses the Wasserstein dependence coefficient in terms of trace operations on covariance matrices, significantly reducing the computational complexity.

Given two random vectors $X\in R^{p}$ and $Y\in R^{q}$ with covariance matrices $\Sigma_{X}\in R^{p\times p}$ and $\Sigma_{Y}\in R^{q\times q}$, respectively, let $\Sigma \in R^{d\times d}$ be the joint covariance matrix, where d=p+q. Denote $\lambda_{j,X}$ and $\lambda_{j,Y}$ as the eigenvalues of $\Sigma_{X}$ and $\Sigma_{Y}$, respectively, and $\lambda_{j}$ as the eigenvalues of the joint covariance matrix Σ, all in descending order. The Wasserstein dependence coefficient $D_{1}\left(\Sigma \right)$ is then given by$$D_{1}\left(\Sigma \right)=\frac{\sum_{j=1}^{p}\lambda_{j,X}^{1/2}+\sum_{j=1}^{q}\lambda_{j,Y}^{1/2}-\sum_{j=1}^{d}\lambda_{j}^{1/2}}{\sum_{j=1}^{p}\lambda_{j,X}^{1/2}+\sum_{j=1}^{q}\lambda_{j,Y}^{1/2}-\sum_{j=1}^{p\vee q}{\left(\lambda_{j,X}+\lambda_{j,Y}\right)}^{1/2}},$$
where p∨q denotes max(p,q). Each predictor is scored using $D_{1}\left(\Sigma \right)$ and ranked accordingly.

This formulation relies solely on eigenvalues of the covariance matrices, avoiding direct computation of optimal transport problems. This makes it a highly efficient and scalable method for measuring dependence, particularly suitable for large-scale genomic datasets such as gene expression and multiomics data in LUAD.

### 4.4. Application to Multiomics Feature Selection in LUAD

We apply the Wasserstein dependence measure to select features from CNA, DNA methylation, and mRNA expression data, treating TMB as the response variable. For each genomic feature $X_{i}$, we compute the Wasserstein dependence $D_{1}\left(X_{i},\mathrm{TMB}\right)$ and rank the features accordingly. The top [n/log(n)] features are retained for further analysis, ensuring that only the most informative genes associated with TMB in LUAD are selected.

By utilizing the Wasserstein dependence coefficient, this approach effectively captures nonlinear and high-dimensional dependencies, providing a more flexible and powerful alteration to conventional feature selection methods.

## Figures and Tables

**Figure 1 bioengineering-12-00694-f001:**
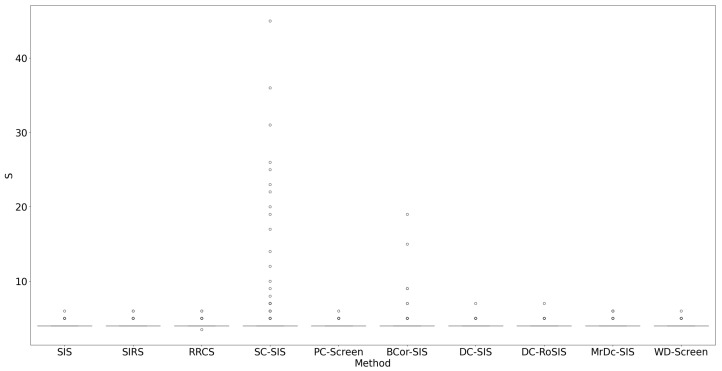
S in Simulation 1. The smaller the S, the better the performance.

**Figure 2 bioengineering-12-00694-f002:**
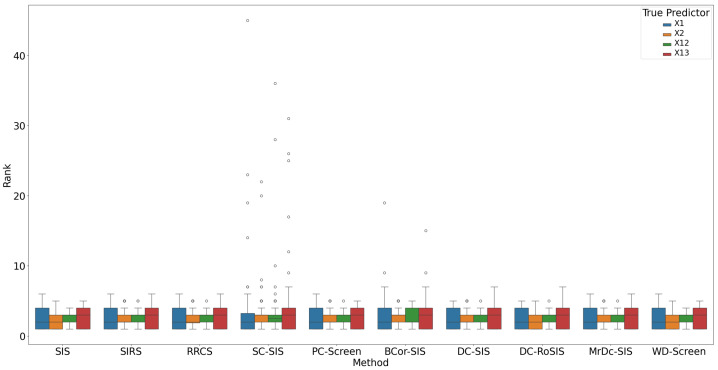
Rank of true predictors in Simulation 1. The smaller the rank, the better the performance.

**Figure 3 bioengineering-12-00694-f003:**
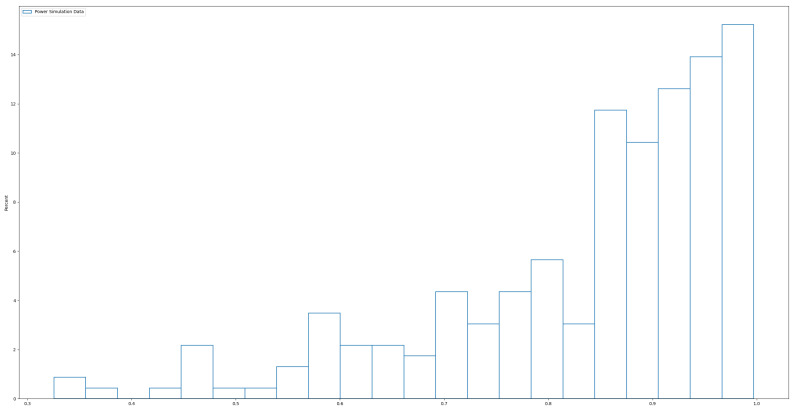
Histogram of simulated power distribution. The simulated data displays a distinct left-skewed distribution.

**Figure 4 bioengineering-12-00694-f004:**
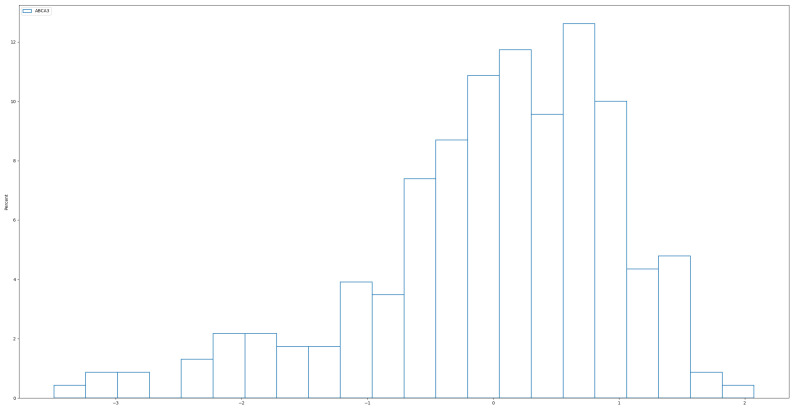
Histogram of real data. The standardized ABCA3 gene expression is also visibly left-skewed.

**Figure 5 bioengineering-12-00694-f005:**
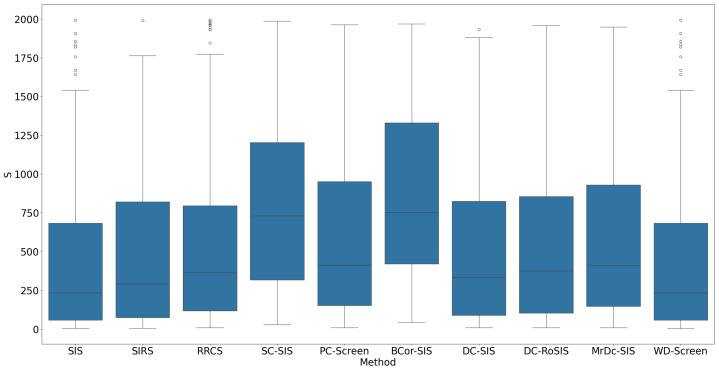
S in Simulation 2. The smaller the S, the better the performance.

**Figure 6 bioengineering-12-00694-f006:**
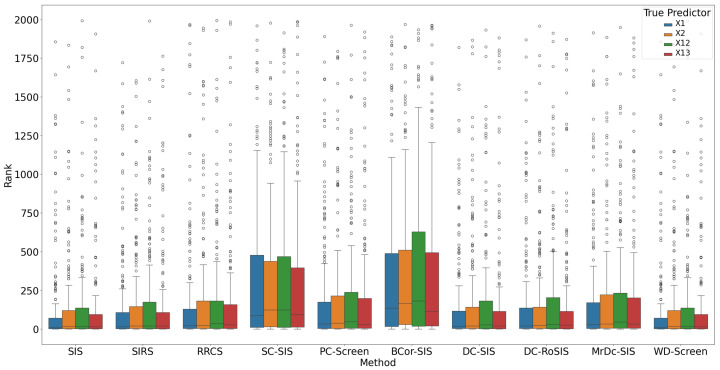
Rank of true predictors in Simulation 2. The smaller the rank, the better the performance.

**Figure 7 bioengineering-12-00694-f007:**
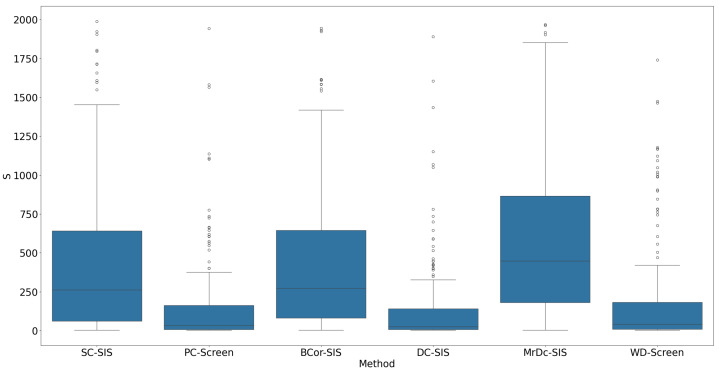
S in Simulation 3. The smaller the S, the better the performance.

**Figure 8 bioengineering-12-00694-f008:**
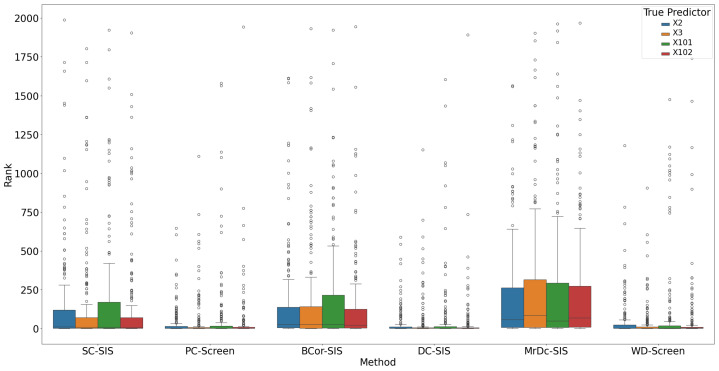
Rank of true predictors in Simulation 3. The smaller the rank, the better the performance.

**Figure 9 bioengineering-12-00694-f009:**
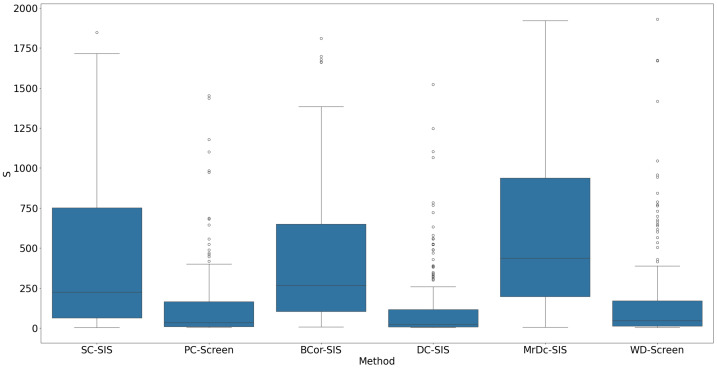
S in Simulation 4. The smaller the S, the better the performance.

**Figure 10 bioengineering-12-00694-f010:**
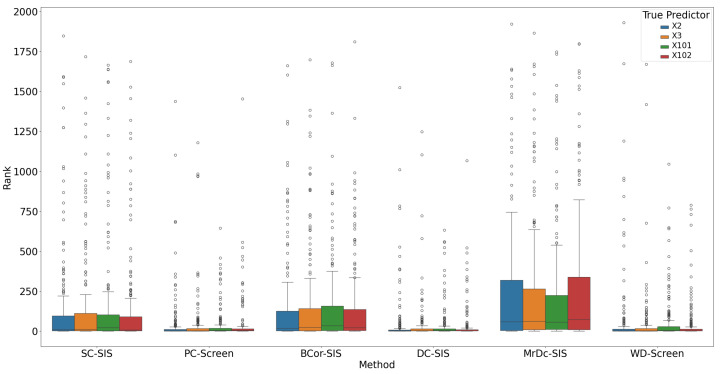
Rank of true predictors in Simulation 4. The smaller the rank, the better the performance.

**Figure 11 bioengineering-12-00694-f011:**
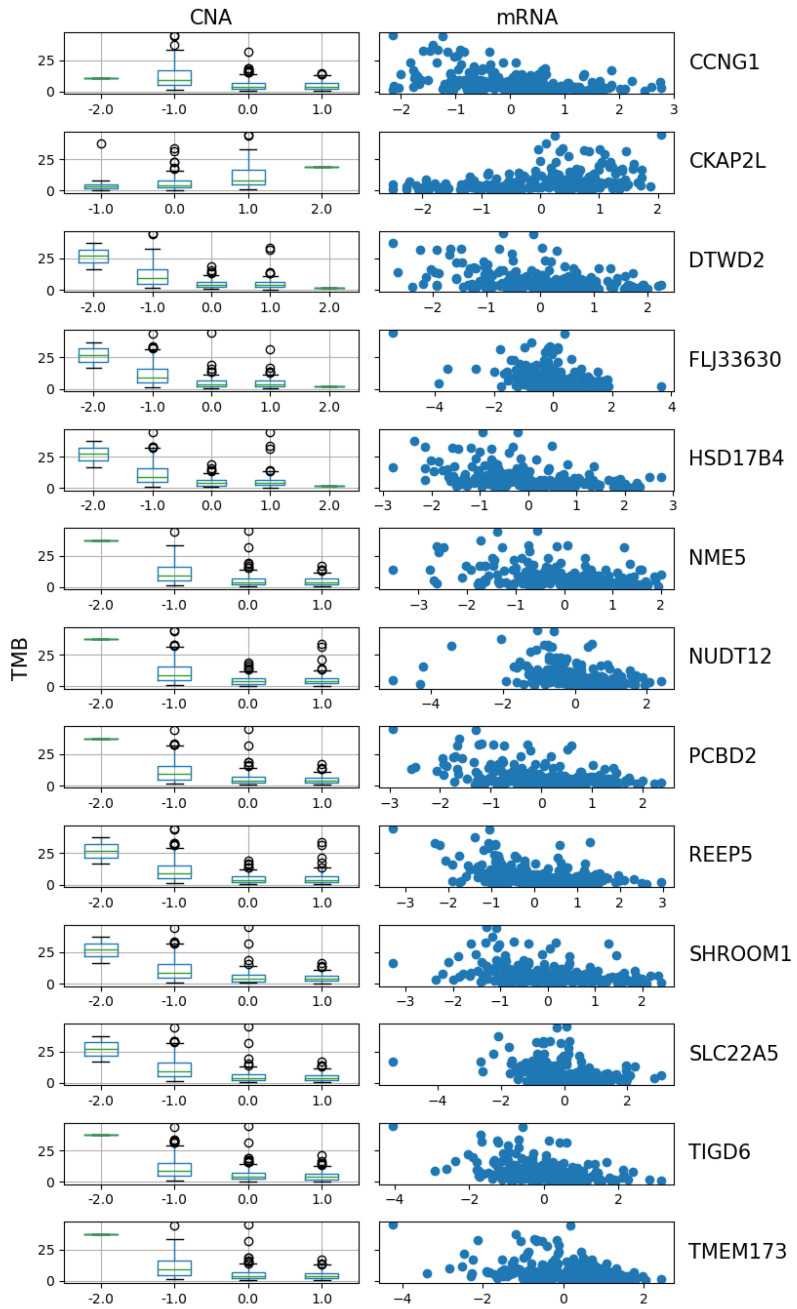
The CNA vs. TMB and mRNA vs. TMB plots of the 13 selected genes.

**Figure 12 bioengineering-12-00694-f012:**
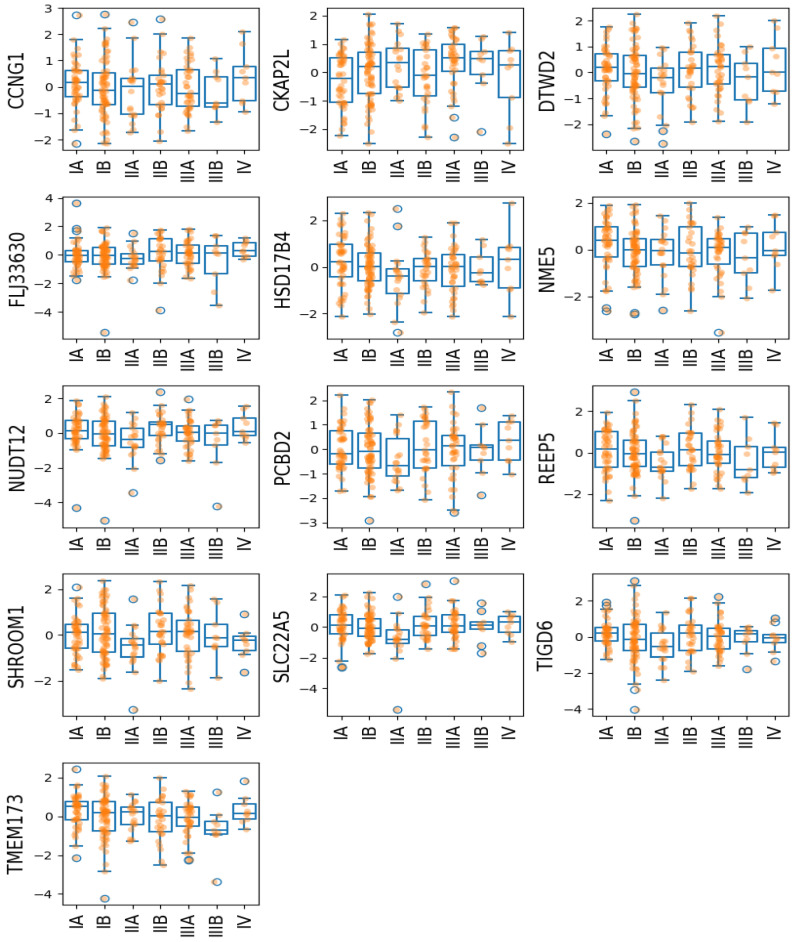
mRNA expression vs. cancer stage of the 13 selected genes. The orange dots are individual mRNA expression values.

**Figure 13 bioengineering-12-00694-f013:**
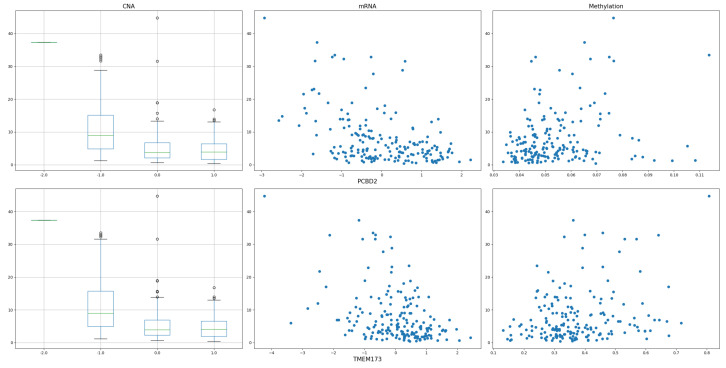
The CNA vs. TMB, mRNA vs. TMB, and DNA methylation vs. TMB of the two common genes.

**Table 1 bioengineering-12-00694-t001:** Summary of simulation studies and real data analyses.

Study	Design	Objective
Sim. Study 1	Multivariate normal predictors with AR(1) covariance; linear response.	Validate method implementations and compare performance in a simple ideal setting.
(Benchmark Test)		
Sim. Study 2	Predictors follow power distribution; response remains linear.	Assess robustness of methods under skewed realistic distributions.
(Non-Gaussian)		
Sim. Study 3	3D predictors (CNA, mRNA, methylation); multivariate response.	Mimic multiomics data structure and test scalability to complex input.
(Multiomics)		
Sim. Study 4	Same structure as Study 3, but with interaction effects in response.	Evaluate methods’ ability to detect interactive non-marginal effects.
(Interactions)		
Real Data 1	TCGA-LUAD: CNA and mRNA; TMB as response.	Identify genes linked to TMB using top three screening methods.
(Two-Platform)		
Real Data 2	TCGA-LUAD: CNA, mRNA, and methylation; TMB as response.	Confirm robustness and detect strongest consistent biomarkers.
(Three-Platform)		

**Table 2 bioengineering-12-00694-t002:** Performance of the three approaches for Simulation 1. The individual success rate $P_{s}$, and the overall success rate $P_{a}$ are demonstrated. The predetermined cutoff for $P_{s}$ and $P_{a}$ is s=[n/log(n)]=37.

Method	$P_{X_{1}}$	$P_{X_{2}}$	$P_{X_{12}}$	$P_{X_{13}}$	$P_{a}$	Method	$P_{X_{1}}$	$P_{X_{2}}$	$P_{X_{12}}$	$P_{X_{13}}$	$P_{a}$
SIS	1	1	1	1	1	BCor-SIS	1	1	1	1	1
SIRS	1	1	1	1	1	DC-SIS	1	1	1	1	1
RRCS	1	1	1	1	1	DC-RoSIS	1	1	1	1	1
SC-SIS	0.995	1	1	1	0.995	MrDc-SIS	1	1	1	1	1
PC-Screen	1	1	1	1	1	WD-Screen	1	1	1	1	1

**Table 3 bioengineering-12-00694-t003:** Performance of the three approaches for Simulation 2. The individual success rate $P_{s}$ and the overall success rate $P_{a}$ are demonstrated. The predetermined cutoff for $P_{s}$ and $P_{a}$ is s=[n/log(n)]=37.

Method	$P_{X_{1}}$	$P_{X_{2}}$	$P_{X_{12}}$	$P_{X_{13}}$	$P_{a}$	Method	$P_{X_{1}}$	$P_{X_{2}}$	$P_{X_{12}}$	$P_{X_{13}}$	$P_{a}$
SIS	0.66	0.605	0.6	0.62	0.165	BCor-SIS	0.335	0.295	0.3	0.345	0
SIRS	0.61	0.59	0.57	0.61	0.14	DC-SIS	0.615	0.57	0.535	0.585	0.11
RRCS	0.595	0.56	0.53	0.55	0.1	DC-RoSIS	0.585	0.565	0.53	0.59	0.11
SC-SIS	0.35	0.33	0.36	0.385	0.005	MrDc-SIS	0.53	0.515	0.47	0.52	0.05
PC-Screen	0.53	0.485	0.465	0.54	0.055	WD-Screen	0.66	0.605	0.6	0.62	0.165

**Table 4 bioengineering-12-00694-t004:** Performance of the three approaches for Simulation 3. The individual success rate $P_{s}$, and the overall success rate $P_{a}$ are demonstrated. The predetermined cutoff for $P_{s}$ and $P_{a}$ is s=[n/log(n)]=37.

Method	$P_{X_{2}}$	$P_{X_{3}}$	$P_{X_{101}}$	$P_{X_{102}}$	$P_{a}$
SC-SIS	0.595	0.68	0.65	0.675	0.195
PC-Screen	0.83	0.85	0.815	0.865	0.505
BCor-SIS	0.575	0.56	0.545	0.6	0.14
DC-SIS	0.85	0.86	0.835	0.87	0.555
MrDc-SIS	0.43	0.4	0.47	0.4	0.035
WD-Screen	0.8	0.83	0.8	0.845	0.475

**Table 5 bioengineering-12-00694-t005:** Performance of the three approaches for Simulation 4. The individual success rate $P_{s}$, and the overall success rate $P_{a}$ are demonstrated. The predetermined cutoff for $P_{s}$ and $P_{a}$ is s=[n/log(n)]=37.

Method	$P_{X_{2}}$	$P_{X_{3}}$	$P_{X_{101}}$	$P_{X_{102}}$	$P_{a}$
SC-SIS	0.63	0.64	0.58	0.655	0.17
PC-Screen	0.83	0.82	0.825	0.88	0.51
BCor-SIS	0.58	0.57	0.52	0.55	0.09
DC-SIS	0.855	0.87	0.865	0.89	0.575
MrDc-SIS	0.415	0.405	0.46	0.395	0.035
WD-Screen	0.835	0.84	0.76	0.81	0.44

**Table 6 bioengineering-12-00694-t006:** Distribution of cancer stages.

Stage I		Stage II		Stage III		Stage IV
**114**		**47**		**47**		**9**
IA	IB	IIA	IIB	IIIA	IIIB	
46	68	18	29	38	9	9

## Data Availability

The TCGA-LUAD data used in this manuscript can be accessed through the following link: https://www.cbioportal.org/study/summary?id=luad_tcga_pub (accessed on 7 June 2025).

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
