# Peer review of "Detection of LUAD-Associated Genes Using Wasserstein Distance in Multiomics Feature Selection"

_bioengineering, 2025, doi:10.3390/bioengineering12070694_

Round 1
Reviewer 1 Report
Comments and Suggestions for Authors
General comments:
- All the figures should be completely redrawn, with larger font sizes and considerably brighter and more contrasting colors. The figures in the current version are almost impossible to read.
- The authors should explain in clearer and more explicit words, what values in the ML study were used as features, and what values/effects/events/classes were predicted. This is particularly important for the computational experiments with synthetic data, but also crucial for the studies of LUAD multi-omics profiles.
Specific comments:
- All acronyms like AR (line 113), i.i.d. (line 138), etc., should be explained in the text. Please also provide an acronym list in the end of the manuscript.
- The introduction section should explain the essentials of the Wasserstein distance methods before it is described in detail in the Methods section.
- Section 2.1 requires a table or a figure that shows the study design.
- Tables 1-4 contain success rates as performance criteria. Could the author provide alternative metrics used for ML quality assessment, such as sensitivity/recall, specificity, precision, MCC, F1, etc.?
- The Materials and Methods section contains the Wasserstein distance method only. However, in the Results section, they apply many methods (such as SIS, SRIS, RRCS, etc.). Please describe in the Materials and Methods section. Please also make references to the code implementation of these methods.
Author Response
We sincerely thank the reviewers for taking the time to carefully evaluate our manuscript. In the attached word document, we provide detailed, item-by-item responses to each of the comments and questions raised.

Reviewer 2 Report
Comments and Suggestions for Authors
The authors describe multi omics analysis of lung adenocarcinoma data. Overall, it is new methodical application.
The text has redundant abbreviations and has not cited terms.
Line 7: “PC-Screen, DC-SIS, and WD-Screen’ – these methods should be commented, the abbreviations shown in full.
Line 11: “via cBioPortal” – need either cite cBioPortal, or not mention it in the Abstract providing reference in the main text.
Line 36: “several studies have focused on multi-omics” – need cite these studies, add references to this phrase.
Line 57: “Projection based Sure Independence Screening (PC-Screen)” – the abbreviation is not understandable – how word ‘PC’ appeared?
Line 62: ‘TCGA-LUAD dataset’ – need give TCGA abbreviation in full, add online-link to this database.
Line 106: ‘ball correlation’ – please comment what is ball here?
Line 111: ‘used in previous papers[ 3 ,4 ,7 ].’ – too many previous papers. Is it replication only? Need show novelty.
Line 113: ‘AR (1)’ – what is abbreviation AR here?
Figures 1 and 2 are too small. Make it on page width.
The figure legend should be clear – Wording “in Study 1” looks like a reference – add wording like ‘in this model Study”
Line 243: ‘2-platform study’ – please change digit to word in the section title (like ‘Two-platform…’)
Same remark is for title 3-platform.
Make all the figures (from 3 to 10) larger, by page width. Every figure is too small, not readable.
Figure 11 also could be rearranged to have larger panels.
References:
Ref.1 – no authors’ names – it is not correct, or typo.
Ref.14 – no page numbers.
Author Response

(The authors gave the same response as above.)

Round 2
Reviewer 2 Report
Comments and Suggestions for Authors
Thanks for the manuscript update and detailed answer. I have no more critical remarks.
Just technical notes:
Line 25 - keywords. Change ' LUAD' to full text in the keywords list.
In the formatting after the formulas (lines 487,490,500) - phrases starting from "where..." should be aligned to left, without indent, as it whole sentence. See correct formatting on line 514.
The ref.1 has old access date (2024). Please refresh.